# A Novel Viscoelastic Deformation Mechanism Uncovered during Vickers Hardness Study of Bone

**DOI:** 10.3390/jfb15040087

**Published:** 2024-03-31

**Authors:** Ahmed Ibrahim, Zhenting Jiang, Khosro Shirvani, Alireza Dalili, Z. Abdel Hamid

**Affiliations:** 1Mechanical Engineering Department, Farmingdale State College, Farmingdale, NY 11735, USA; khosro.shirvani@farmingdale.edu (K.S.); dalilia@farmingdale.edu (A.D.); 2The Department of Earth & Planetary Sciences, Yale University, New Haven, CT 06511, USA; zhenting.jiang@yale.edu; 3Central Metallurgical Research and Development Institute, Helwan 11421, Egypt; forzeinab@yahoo.com

**Keywords:** Vickers microhardness of bone, viscoelastic deformation of bone, bone quality, “screw-like” deformation mechanism, mechanical properties of bone, nanoindentation of bone, bone hardness, bone microstructure, anisotropic properties of bone

## Abstract

This study investigates the viscoelastic deformation mechanisms of bone as a response to Vickers hardness indentation. We utilized advanced high-resolution scanning electron microscopy (SEM) to investigate a distinct deformation pattern that originates from the indentation site within the bone matrix. The focus of our research was to analyze a unique deformation mechanism observed in bone tissue, which has been colloquially termed as “screw-like” due to its resemblance to a screw thread when viewed under an optical microscope. The primary goals of this research are to investigate the distinctive characteristics of the “screw-like” deformation pattern and to determine how the microstructure of bone influences the initiation and control of this mechanism. These patterns, emerging during the dwell period of indentation, underscore the viscoelastic nature of bone, indicating its propensity for energy dissipation and microstructural reconfiguration under load. This study uncovered a direct correlation between the length of the “screw-like” deformation and the duration of the indentation dwell time, providing quantifiable evidence of the bone’s viscoelastic behavior. This finding is pivotal in understanding the mechanical properties of bone, including its fracture toughness, as it relates to the complex interplay of factors such as energy dissipation, microstructural reinforcement, and stress distribution. Furthermore, this study discusses the implications of viscoelastic properties on the bone’s ability to resist mechanical challenges, underscoring the significance of viscoelasticity in bone research.

## 1. Introduction

Bone, a highly sophisticated and organized tissue, serves as the foundational structural framework of the vertebrate body, characterized by its hierarchical structure that ranges from the macroscopic level to the nanoscale. This tissue comprises a balanced mix of inorganic and organic constituents, with 50 to 70% mineral content, predominantly hydroxyapatite, which confers strength and stiffness, essential for supporting weight and enduring mechanical forces. Additionally, from 20 to 30% of bone’s makeup is an organic matrix, primarily Type I collagen, providing the necessary flexibility and toughness for impact absorption without fracturing [1]. Bone tissue is classified into two unique types, distinguished by their density and porosity: cortical (compact) bone and trabecular (cancellous or spongy) bone. Dense cortical bone forms the rigid exterior of long bones, offering protection over the more porous trabecular bone. This latter type, located at the ends of long bones and within vertebrae, features a spongy, lattice-like structure optimized for impact absorption.

Bone, a critical component of the human musculoskeletal system, is not just a static structural element but a dynamic tissue that constantly remodels and reacts to mechanical stimuli. Its mechanical properties, including hardness, density, and elasticity, are essential in maintaining the body’s structural integrity, facilitating movement, and protecting vital organs [2]. Studying these properties is fundamental to understand bone physiology and crucial in clinical contexts, particularly in diagnosing and treating bone diseases and designing orthopedic implants [3].

Historically, the mechanical characterization of bone has predominantly focused on its elastic properties, with less emphasis on its viscoelastic behavior, especially under sustained loads [4]. Traditional models often treat bone as a homogenous, isotropic material, simplifying its complex, heterogeneous nature [5]. However, such models are limited in their ability to accurately predict bone’s behavior under various loading conditions, particularly over extended periods. This knowledge gap becomes particularly relevant in understanding the progression of stress-related injuries and the performance of bone-anchoring medical devices over time. The Vickers hardness test, known for its precision in measuring the resistance of materials to deformation, offers a unique opportunity to explore these aspects of bone mechanics [6]. Applying a known load for a specific duration can elucidate the relationship between load duration and bone deformation, shedding light on the time-dependent aspects of bone’s mechanical response [7].

This study aims to bridge the gap in understanding the time-dependent deformation of bone. The focus is on how bone’s response to mechanical stress varies with the duration of load application, as measured by the Vickers hardness test [8]. Investigating this relationship is vital for several reasons. It has implications for the design and longevity of orthopedic implants, which must withstand varying loads over time. From a physiological perspective, understanding the viscoelastic properties of bone can enhance the diagnosis and management of conditions like osteoporosis, wherein altered mechanical properties increase fracture risk [9]. Additionally, this research has potential applications in biomechanical modeling, contributing to more accurate predictions of bone behavior under different stress conditions. By elucidating these aspects of bone mechanics, this study aims to contribute valuable insights to biomechanics, orthopedics, and material science, ultimately improving patient care and advancing our understanding of bone physiology [10].

In exploring the viscoelastic deformation mechanisms of bone, this study investigates the bone’s response to the Vickers hardness test, a precise method of applying sharp contact loading that is essential for determining the initiation and development of fractures. Utilizing high-resolution scanning electron microscopy (SEM), we have analyzed the intricate deformation patterns that emerge within the bone matrix, specifically focusing on “screw-like” separations and ridges. These features, which manifest during the indentation’s dwell time, reveal the viscoelastic nature of bone, characterized by its ability to dissipate energy and undergo microstructural reconfiguration when subjected to mechanical stress. A notable discovery of our research is the direct correlation between the “screw-like” deformation length and the dwell time of indentation, providing critical insights into the viscoelastic properties and fracture toughness of bone. This correlation illuminates the complex interplay between energy dissipation, microstructural reinforcement, and stress distribution within bone, emphasizing the profound implications of viscoelasticity in bone mechanics and its significance in medical applications. This deformation pattern marks a novel finding, undocumented in earlier research. It was first identified by A. Ibrahim, the lead researcher of this study, during a comprehensive sequence of Vickers hardness tests on bone in the summer of 2022.

## 2. Biological Significance of Bone Hardness

The biological significance of bone hardness is rooted in the material’s role in the structure and function of the skeletal system. Bone hardness, a measure of resistance to deformation and indentation, is a critical factor that reflects bone tissue’s health and mechanical competence [11]. It is primarily determined by the mineral content—mainly hydroxyapatite—within the organic matrix of collagen fibers, which together provide a balance of rigidity and toughness [12].

From a physiological perspective, the hardness of bone is essential for maintaining proper support for the body, protecting internal organs, and offering a rigid attachment for muscles, thus enabling locomotion [13]. The ability of bone to resist deformation is not only a determinant of skeletal integrity but also a crucial factor in preventing fractures [14]. Bones with higher hardness are generally more resistant to wear and less prone to damage under normal daily loads, a property that is especially important in weight-bearing bones such as the femur and spine [15].

On the other hand, the variability in bone hardness across different locations and ages reflects the adaptive nature of bone to various mechanical demands [16]. This adaptive mechanism, known as Wolff’s Law, states that bone density and structure will adapt to the loads under which it is placed [17]. Therefore, bone hardness is a dynamic property influenced by age, nutrition, physical activity, and overall health [18].

In a clinical setting, understanding bone hardness has implications for diagnosing and treating bone diseases like osteoporosis, where decreased bone hardness and density lead to an increased risk of fractures [19]. Assessing bone hardness can help detect and monitor a treatment’s effectiveness early [20]. Furthermore, bone hardness knowledge in biomedical engineering is crucial for designing orthopedic implants and prosthetics [21]. These devices must match the mechanical properties of bone to avoid issues such as implant loosening or failure.

Thus, the study of bone hardness intersects with various disciplines, from biology and medicine to materials science and biomechanical engineering, highlighting its broad significance in both fundamental science and applied health sciences.

## 3. Relevance of Vickers Hardness Test in Biomaterials

The relevance of the Vickers hardness test in the study of biomaterials, particularly bone, is profound due to its precision and applicability in quantifying material properties. The Vickers hardness is an ASTM International standard test, ASTM E384 [22]. This test, characterized by applying a diamond pyramid indenter onto the material’s surface, provides a measure of hardness defined as the material’s resistance to plastic deformation. This metric is particularly valuable in assessing the mechanical properties of biomaterials like bone, dental enamel, and engineered tissues, which require a delicate balance of strength and resilience.

Bone hardness, encapsulating its ability to undergo both elastic and plastic deformation, is integral to the structural and functional competency of the skeletal system. Hardness, a descriptor of material resistance to indentation, is particularly significant in engineering materials where it has traditionally held a reliable correlation with strength. Typically, for metals, the Vickers hardness number (Hv) is empirically approximated to be three times the yield strength (σy) [22]. This relationship, however, is not exact for biological materials such as bone.

The technique of microhardness testing utilizes a diamond indenter to apply a force that produces an indentation on the material’s surface. The Vickers hardness number is then determined from the indentation force and the indentation area, as demonstrated in Figure 1. The Vickers hardness test, employing a diamond pyramid indenter, is a standard method for measuring the hardness of materials, including biological materials at the micro and nano scales, the latter being assessed by nanoindentation [23,24]. Nanoindentation not only measures hardness but also evaluates the elastic modulus at the nanoscale level, pertinent to the lamellar structure of bone [25].

Bone hardness has been found to strongly correlate with mineralization, a key factor in its mechanical properties [26]. This relationship has been demonstrated in studies measuring mechanical property variations within human vertebrae, where nanoindentation revealed significant differences in hardness between anterior and posterior regions, likely due to the differential degree of mineralization heterogeneity.

Indentation techniques have gained prominence in evaluating the mechanical integrity of biomaterials and hard tissues, including their fracture toughness and strength [27]. These methods are invaluable in materials science, contributing to the advancement of biomimetic materials and providing detailed insights into the mechanical behavior of bone at microscopic levels.

## 4. Implications of Time-Dependent Deformation

The implications of time-dependent deformation in materials, particularly in biological tissues like bone, are profound and diverse. This phenomenon, characterized by a change in deformation relative to the duration of applied load, underscores the viscoelastic properties of bone [28,29,30,31]. Such materials exhibit viscous and elastic behaviors under stress, meaning bone’s response to loading is inherently time-dependent, with its load-bearing capacity and deformation varying with the nature and duration of the stress.

In biomedical engineering, grasping time-dependent deformation is essential. For orthopedic surgery and prosthetic design, it is critical to account for bone’s viscoelasticity to ensure the durability and compatibility of implants and prosthetics. Insights into time-dependent deformation can help mitigate issues like stress shielding, a common complication leading to bone resorption when implants bear excessive loads.

This knowledge is equally crucial for understanding and managing osteoporosis and other bone disorders, which affect bone’s microarchitecture and mechanical properties, thereby influencing its time-dependent deformation behavior [32,33,34,35]. Comprehending how diseased bone responds to sustained stress is indispensable for developing effective therapeutic and preventative fracture strategies [33].

In material science and biomechanics, the investigation into time-dependent deformation challenges the traditional elastic or perfectly plastic paradigms, suggesting the need for more intricate models to predict bone behavior under physiological loads [36] accurately. These models intended for interaction with the human skeletal system can significantly enhance the design and testing of biomaterials.

Overall, studying time-dependent deformation in bone paves the way for innovative research and applications across the medical, biomechanical, and material sciences fields. It emphasizes the complexity of biological tissues and the necessity for sophisticated approaches in their analysis and clinical use.

## 5. Materials and Methods

Tibia bones of white-tailed deer were obtained from a local processing factory. All soft tissues were removed, and the bones were stored at −20 °C. Before testing, the bones were soaked in a 3% hydrogen peroxide (H_2_O_2_) solution for 24 h. The cross-section of the samples was prepared following ASTM E384. First, the bone’s main shaft (diaphysis) was sectioned into 0.250-inch-thick segments using a Buehler-ISOMET 4000 (Buehler Inc., Lake Blu, IL, USA) precision saw equipped with a diamond blade. The cleaned cross-section was then cold-mounted using epoxy resin. After mounting, the samples were subjected to automatic grinding using a Buehler-AutoMet Grinder-Polisher (Buehler Inc., Lake Blu, IL, USA) in steps (180, 240, 800, and 1200 grit). Each specimen was meticulously inspected under a microscope to ensure the absence of pre-existing microdamage, which could influence the hardness test results.

Vickers Hardness Testing Protocol: The Vickers hardness test was conducted using a microhardness tester with a diamond pyramid indenter (HM123) (Mitutoyo Corp, Aurora, IL, USA) with a 50 g load and a 15 s dwell time. Measurements were taken in Vickers hardness number (VHN) in accordance with ASTM E384 [22]. Every measurement recorded was an average of three measurements. A minimum distance of 3d (diagonal of indentation) was kept between any two consecutive indentations.

A systematic approach was adopted wherein each bone sample underwent a series of indentations under controlled conditions. The primary variable in this study was the dwell time—the duration for which the indenter was kept under maximum load on the bone surface. A range of dwell times was selected to investigate the time-dependency of deformation. The load applied was kept constant across all tests to isolate the effect of dwell time.

In this study, the Nikon LV-150N Microscope, outfitted with a high-resolution digital camera, played a crucial role in capturing the macroscopic images essential for our analysis. Its advanced optical system allowed for exceptionally clear visualization of the indentation marks and the bone matrix, which was fundamental for the initial examination of the deformation caused by the Vickers hardness test as seen in Figure 2. Furthermore, this microscope’s capabilities were particularly evident in the vivid depiction of the “screw-like deformation” displayed in Figure 3, where the precise features of this unique pattern were brought to light, providing a critical foundation for the subsequent, more detailed microscopic analysis.

The SEM Hitachi SU7000, a high-performance scanning electron microscope, was employed for the detailed microscopic analysis. Its superior resolution facilitated a deeper dive into the “screw-like” deformations and the associated microstructural features, such as the ridges and interlamellar spaces. The high magnification and depth of field of the SU7000 made it possible to observe the unique type of plastic deformation characterized by ridges, which are indicative of tearing deformation across the lamellar layers, thus providing invaluable insights into the viscoelastic response of bone to mechanical stress. The combination of these sophisticated imaging tools has been pivotal in advancing our understanding of the complex deformation mechanisms of bone under Vickers hardness testing.

### Declaration of Generative AI and AI-Assisted Technologies in the Writing Process

While preparing this work, the authors used ChatGPT and Grammarly-AI as writing assistants to refine the manuscript’s language and enhance clarity. After using these tools, the authors reviewed and edited the content as needed and took full responsibility for the publication’s content. We did not use Generative AI or AI-assisted tools to create or alter images submitted in this manuscript.

## 6. Results and Discussion

Figure 2 shows a cross-sectional view of a deer tibia bone, where specific sites (indicated by red dots) have been subjected to Vickers hardness testing. This systematic approach to testing allows for a focused analysis of the bone’s response to indentation in distinct, localized areas. Repeated tests in these regions have resulted in a consistent “screw-like” deformation pattern, which provides significant insights into the bone’s mechanical behavior under repeated stress, as seen in Figure 2b. The directionality of these deformations, captured consistently across multiple tests, underscores a directional dependency in the bone’s response, which could be attributed to the intrinsic anisotropic nature of its microstructure.

Figure 3 presents a magnified optical microscope view, showcasing the detailed “screw-like” deformation resembling a screw thread. This specific and repeated deformation pattern seen under high magnification offers a window into the bone’s complex viscoelastic properties. The distinct ridges and grooves that form this “screw-like” pattern illuminate the interplay between the bone’s microarchitecture—namely, the alignment of mineralized collagen fibers—and its macroscopic mechanical resilience. Such patterns are not just random artifacts but rather deliberate responses to mechanical stress, revealing how the bone’s internal structure, down to the microscopic level, plays a pivotal role in its ability to handle and dissipate the forces it encounters, which is crucial for understanding fracture mechanics and the development of tough, biomimetic materials. The “screw-like” deformation mechanism occurred during the dwell time, which is the period when the indenter is held at the maximum load before being retracted.

### 6.1. Screw-like Deformation in Bone during Indentation

The presence of a “screw-like” deformation mechanism on the bone’s surface is truly captivating. This kind of deformation has not been previously documented in the literature. This deformation could result from various factors, including the structural properties of the bone, the duration of the dwell time, and the microstructural response of bone tissue to indentation.

The micrograph at 500× magnification reveals a highly detailed view of a unique “screw-like” deformation in the bone, occurring during the indentation’s dwell time. The image showcases the deformation with exceptional clarity, highlighting its intricate spiral pattern. The total length of this feature extends to 250 μm, with a fine thickness of merely 8 μm. Such precise measurements are crucial in understanding the scale of the deformation, illustrating the microscopic level at which these changes in the bone structure have taken place.

The deformation patterns observed suggest that the bone’s internal structure influences the viscoelastic response, such as the mineralized collagen fibers’ orientation and the mineral matrix’s distribution. The “screw-like” pattern could be aligned with the direction of collagen fibers, affecting how the material deforms under load.

Observing such features can be important for understanding bone behavior under load and for applications such as orthopedic implant design, where materials are often required to mimic or interact with bone tissue. It is also valuable in materials science for studying the mechanical properties of biological materials.

The appearance of a “screw-like” pattern in the bone following a Vickers hardness test is an unusual and intriguing observation that suggests several essential implications in the fields of materials science, biomechanics, and medical research:

Bone Microstructure: The deformation patterns observed in Figure 2, which are oriented in a specific direction, indeed hint at the influence of the bone’s internal structure on its viscoelastic response. The directionality of these patterns could be closely associated with the orientation of mineralized collagen fibers and the distribution of the mineral matrix within the bone. Collagen fibers provide tensile strength and contribute to the bone’s toughness and flexibility, while the mineral matrix offers compressive strength. Their collective arrangement within the bone likely dictates how it deforms under indentation stress. This relationship between structure and mechanical response is fundamental to biomechanics and has significant implications for understanding bone behavior under normal physiological conditions and in response to injury.

Implant Material Design: Understanding the deformation behavior of bone at a microscopic level is crucial for designing biomedical implants. The response of bone tissue to mechanical loads must be considered to ensure the implant’s compatibility and longevity.

Biomechanical Models: This finding may help refine biomechanical models of bone tissue, which could lead to better predictions of fracture risk and the development of preventive strategies against bone-related diseases.

The novelty of this observation suggests a gap in the current literature regarding the behavior of bone under indentation. This could be an opportunity for original research and publication. The significance of such a finding would likely prompt a detailed investigation, including repeating the test under various conditions and examining the bone’s microstructure through other methods, such as computed tomography (CT) scans, to understand the underlying mechanisms at play.

### 6.2. Observation of Time-Dependent Deformation in Bone during Vickers Hardness Testing

A notable time-dependent deformation pattern emerged when conducting a series of Vickers hardness tests on bone tissue. The tests revealed a “screw-like” deformation originated from the indentation perimeter, with its length directly correlating with the indentation dwell time. Specifically, when the dwell time was doubled, the length of this separation also nearly doubled, suggesting a linear relationship between the time under load and the extent of the deformation. This observation indicates that the bone’s response to the applied load is not merely elastic but also viscoelastic, with the deformation being significantly influenced by the duration of load application. The reproducibility of this pattern across multiple tests underscores its potential importance in understanding the mechanical behavior of bone under sustained stresses (Figure 2). This time-dependent deformation behavior could have significant implications for both the biomechanics of bone under physiological conditions and the design of orthopedic implants wherein sustained load bearing is required.

The observation that the length of the “screw-like” deformation in the bone is dwell time-dependent, with its extent nearly doubling when the dwell time of the indentation test is doubled, is a remarkable finding that has several significant implications:

Dwell Time Sensitivity: The result indicates that bone material is highly sensitive to the duration of load application. This suggests that the viscoelastic properties of the bone—where the stress response is time-dependent—influence the deformation patterns. Bone tissue exhibits viscoelastic behavior, meaning it deforms differently under different load application rates and durations.

Deformation Mechanism: The time-dependent nature of the deformation could be linked to the creep behavior of bone, which is the tendency of a solid material to move or deform permanently under the influence of mechanical stresses. This could suggest that the separation is a form of microscale creep deformation occurring within the bone due to the prolonged application of the indenter.

Stress Relaxation: Another aspect could be stress relaxation in the bone material, where the stress within the material decreases under constant strain over time. This relaxation could lead to an increase in the length of the deformation as the material under the indenter yields more over time.

Fracture Propagation: The increased length of the “screw-like” deformation could indicate that microfractures are propagating more with longer loading times. This would be consistent with the slow growth of fractures in brittle materials like the mineral phase of bone.

Clinical Relevance: This time dependency could have clinical relevance. It suggests that the microscale response of bone to sustained loads is essential to consider when assessing the risk of stress fractures.

### 6.3. Understanding the Screw-like Deformation Induced during Indentation

Figure 4 is a high-resolution scanning electron microscope (SEM) image showing the feature described previously as a “screw-like” deformation seen at higher magnification. This feature is of particular interest because it provides a close-up view of the microstructural changes in the bone due to mechanical stress or an applied load, from a Vickers hardness test. This “screw-like” feature may represent a form of localized shear or twisting deformation within the bone matrix. This spiral or helical deformation suggests a complex interaction between the bone’s organic components, mainly mineralized collagen fibers, and its inorganic mineral phase. Such patterns could arise when the applied load causes the collagen matrix to unravel or shear in a direction that reflects its natural orientation, possibly indicating the direction of mineralized collagen fiber alignment in the bone.

The “screw-like” deformation feature, originating from the diamond indentation and continuing on the surface, generated during the dwell time of a Vickers hardness test, is a particularly intriguing observation. This observed pattern reveals a distinct type of plastic deformation in the bone matrix that has not been identified in prior hardness assessments of bone. Therefore, studying such deformation patterns can provide valuable insights into the fundamental biomechanical behavior of bone and inform various applications, from clinical treatments to developing bioinspired materials.


**The “screw-like” pattern implies several things:**
Plastic Deformation: The bone material has undergone plastic deformation, which has caused permanent changes in the microstructure. The continuation of the deformation pattern on the surface of the bone, beyond the immediate area of indentation, indicates that the effects of the applied force during testing are not confined to just the point of contact where the indenter is pressed into the bone material. Instead, the deformation has propagated beyond the initial contact zone, affecting a larger area of the bone’s surface. The spread of deformation could provide insights into the microstructural properties of the bone, such as the distribution and bonding of the mineral and organic components. It may indicate a certain level of toughness and the ability to dissipate energy from the point of impact. The observation could also be related to the structural integrity and homogeneity of the bone.Time-Dependent Deformation: Since this feature was generated during the dwell time, it suggests that the bone’s response to the applied stress is time-dependent. This is characteristic of viscoelastic materials, where the loading rate and the applied load’s duration significantly influence the deformation behavior.Anisotropy in Bone: The “screw-like” deformation’s directional nature indicates anisotropy in the bone’s mechanical properties. Bone is a composite material with a complex internal structure, including mineralized collagen fiber orientation (CFO) and mineral content distribution. Therefore, when considering the microstructural behavior of bone under compression and the resulting tensile stresses from an indentation test, it is logical to deduce that the fracture, such as the “screw-like” deformation, would form in a direction that is perpendicular to the CFO. This alignment exploits the anisotropic nature of bone and its varying resistance to stress in different directions, following the path of least resistance as predicted by conventional principles.The deformation pattern suggests that the force exerted by the indenter may have been aligned with the direction of these internal structures, leading to a more pronounced deformation in that particular direction.Microstructural Adaptation: The deformation pattern could be a microstructural adaptation, where the bone’s internal structure allows it to deform in a specific way to distribute the stress and minimize damage. This can be an adaptive quality of bone to handle stresses and strains encountered during normal physiological activities.Implications for Bone Health and Disease: Observing such deformation patterns can provide insights into the health of the bone. In pathological states like osteoporosis, the bone matrix may deform differently due to mineral density and collagen cross-linking alterations, affecting the material properties and deformation patterns.


This unique deformation characteristic captured by high-resolution SEM imaging is fascinating from a materials science perspective. It has significant implications for understanding bone behavior, designing biomedical devices, and potentially diagnosing and treating bone-related diseases. Further investigation into this pattern could yield important information about the mechanical behavior of bone under different loading conditions and over time.

### 6.4. Understanding the Ridge Pattern Deformation

The intricate patterns of ridges seen in the SEM image following a Vickers hardness test highlight the nuanced reaction of bone to applied stress. These ridges are not merely surface irregularities; they tell a story of the bone’s resistance and adaptation to the mechanical load.

Firstly, these ridges could be manifestations of the bone undergoing plastic deformation. When the applied stress from the indenter surpasses the bone’s elastic threshold, it results in permanent deformation, characterized by these pronounced lines. They potentially map out the pathways of stress distribution, revealing the direction in which the bone matrix has been forced to adapt and shift.

Secondly, the ridges may signify the displacement of bone material during the indentation process. As the indenter applies pressure, it forces the organic and inorganic constituents of the bone to move. During the dwell phase, this pressure is constant, causing the material to migrate and form ridges that mark the movement’s flow. This process underscores the dynamic nature of the bone’s response under continuous load, adapting its structure in real time.

Lastly, the arrangement of the ridges could correlate with the orientation of mineralized collagen fibers within the bone. These fibers, crucial for the bone’s tensile strength, may become more apparent under the stress of indentation. The alignment and response of these fibers under load are key to understanding the bone’s capacity for energy absorption and dissipation. Furthermore, the presence of these ridges accentuates the inherent anisotropic qualities of bone—the variation in its mechanical properties based on direction. This anisotropy stems from the complex hierarchical organization of bone, from the molecular composition to the larger-scale structure of osteons and trabeculae, all contributing to its ability to handle diverse mechanical challenges.

Understanding the precise nature of these ridges and the conditions under which they form can provide significant insights into the mechanical behavior of bone, the interaction of its composite structures under load, and the potential for designing biomimetic materials and implants that can better integrate with the natural biomechanics of bone tissue. Further analysis using different imaging techniques, or even mechanical testing at different scales, could provide more information about the significance of these ridges in the context of bone mechanics.


**Ridge Pattern Deformation as a Tearing and Shear**


The ridges observed in the SEM image, indicative of tearing and shear deformation through the layers of lamellae, suggest a multi-step deformation process within the bone’s microstructure. This type of deformation reflects the complex biomechanical behavior of bone under loading conditions imposed by the Vickers hardness test. Here is an in-depth exploration: Shear Stress Mechanism: The ridges are likely the result of shear stresses that occur when the applied force is not perpendicular to the surface. As the diamond indenter penetrates the bone, it induces both compressive and shear stresses. These shear stresses can displace material laterally, creating ridges as the bone matrix tears and deforms.Lamellar Tearing: The step-by-step tearing through the lamellae suggests a lamellar tearing mechanism, where the layers of bone are sequentially separated. This indicates that the adhesion between the lamellae was overcome by the shear forces, which could be related to the organic components of the bone matrix, such as the mineralized collagen fibers, being disrupted.Interlamellar Weakness: The interlamellar spaces may represent zones of weakness within the bone’s microstructure. These zones could be more susceptible to deformation due to differences in material composition and mechanical properties between the mineralized matrix and the organic fibers.Microcrack Propagation: The ridges may also be paths of microcrack propagation. Microcracks can initiate at points of stress concentration, such as around osteocyte lacunae or along Haversian canals, and then propagate along the lines of least resistance, which can create the appearance of ridges.Collagen Fiber Orientation: The orientation and distribution of mineralized collagen fibers play a significant role in the propagation of shear deformation. If the indenter force aligns with the direction of the collagen fibers, it could facilitate the separation between lamellae, leading to a more pronounced ridge pattern.Dwell Time Influence: The dwell time during the hardness test is a period of sustained load, allowing for continued deformation, including creep. The longer the load is maintained, the more pronounced the deformation can become, as the bone continues to experience shear and tearing.Anisotropic Response: Bone’s anisotropic nature means it responds differently to stresses depending on the direction relative to its microstructure. The ridges may illustrate this anisotropic response, showing how the bone’s internal architecture dictates the path and pattern of deformation.Biological Implications: These tearing and shear features are not just physical phenomena but have biological implications. The mechanical environment influences cellular activity, and significant deformation can trigger an osteogenic response, potentially leading to bone remodeling and changes in microarchitecture.

Understanding the nature of these ridges and the tearing of lamellae provides valuable insights into the mechanical behavior of bone and its structural integrity. It has direct implications for the fields of orthopedics, particularly in the context of fracture risk assessment, the development of biomaterials that mimic bone’s mechanical properties, and the design of orthopedic implants that are compatible with the bone’s natural load-bearing capabilities.

### 6.5. Role of Microstructure in Controlling Viscoelastic Deformation

Figure 2b illustrates the outcome of a methodical examination of bone subjected to indentation testing in specific, targeted zones. The consistent appearance of “screw-like” deformations after indentation tests on bone samples points to a specific and reproducible response of the bone’s microstructure to mechanical stress. Notably, this deformation pattern does not occur universally across all tests—it is only present in particular samples and specific regions, suggesting that it is highly dependent on the local microstructural characteristics of the bone. Although the exact microstructural features that initiate this unique deformation have not yet been identified, it is clear that the phenomenon is not random but tied to the intricate structure of the bone at a microscopic level.

The deformation’s directionality and repeatability also imply that the bone can dissipate energy from the indentation force in a controlled manner. This ability is vital for bones’ functionality, as they need to withstand varied and repeated loads without failing. The controlled deformation mechanism observed could be a protective response, allowing the bone to absorb impact and distribute stress while minimizing damage. Understanding this mechanism is crucial for designing better biomaterials and orthopedic solutions that can mimic the natural behavior of bone under load. As mentioned before, a time-dependent deformation pattern emerged when conducting a series of Vickers hardness tests on bone tissue. The tests revealed that the “screw-like” deformation length directly correlated with the indentation dwell time.

Figure 5 shows an SEM image encapsulating a dynamic and complex microstructure within bone tissue, characterized by a network of cracks, osteocyte lacunae, and osteonal systems. The presence of two Vickers hardness indentations with similar hardness values and proximal placement within this active microstructural landscape is intriguing, especially with the “screw-like” deformation patterns emanating from each indentation site.

Despite the apparent high activity within this microstructure, which includes multiple features such as osteons and microcracks, the “screw-like” formations show a striking regularity in their direction and length. This uniformity across both indentations, despite being set in a region rich with various microstructural entities, strongly implies a directional preference in the viscoelastic response of the bone to mechanical stress.

The alignment of the deformation paths, as indicated by the red arrows and the overarching white arrow, may be influenced by the orientation of the mineralized collagen fibers or the predominant directional load-bearing capacity of this particular bone area. This consistent deformation behavior, even in a complex and “busy” microstructure, offers a window into the bone’s ability to handle stress in a highly controlled and repeatable manner. The fact that the deformations occur from nearly identical start points and extend to similar lengths could suggest that certain microstructural configurations are particularly prone to energy release via these “screw-like” formations under mechanical loading.

This image contributes valuable insights into bone’s mechanical behavior and intrinsic properties, particularly how it manages and distributes energy under loading conditions that induce plastic deformation. It also reinforces the notion of bone’s anisotropy and its capacity to direct energy dissipation in a way that likely contributes to its toughness and durability.

Figure 6 shows a Vickers indentation in a bone area where three osteons converge—two significantly large—providing a unique insight into how bone microstructure can influence deformation mechanisms. In this scenario, the “screw-like” viscoelastic deformation appears to navigate toward the path of least resistance, avoiding the highly stressed region around the osteons. This avoidance results in a relatively short deformation of 43 microns, which is indicative of the directionality and magnitude of the deformation being influenced by localized stress fields.

Osteons, the fundamental structural units in compact bone, are understood to significantly contribute to bone’s mechanical strength and toughness due to their complex, hierarchical organization. When an indentation occurs across osteons, as observed where 60% overlays osteon 2 and 40% osteon 1, it suggests a differentiated response in the bone’s ability to absorb and distribute the stress of the indentation load. The osteons’ orientation and size could create variations in density and stiffness, thereby dictating the flow of energy and the resulting deformation path.

This microstructural configuration can act as a natural barrier to deformation, diverting the “screw-like” pattern away from areas of high stress concentration, which could otherwise lead to crack initiation and propagation. Such behavior underscores the adaptive nature of bone to mechanical challenges, where the microstructure not only supports the load but also plays a role in the energy dissipation processes, essential for preventing fractures and maintaining bone integrity. Understanding these mechanisms is crucial for replicating bone-like properties in synthetic materials and for evaluating bone health in clinical settings.

Figure 7 shows a Vickers hardness indentation on a region of bone with a relatively low hardness value (38 HV), resulting in a “screw-like” deformation extending approximately 130 microns from the indentation site. This considerable deformation length, which is more than three times longer than in the previously mentioned scenario, suggests that the bone’s capacity to absorb and dissipate energy is related to its microstructural composition.

In contrast to the previous, more congested microstructure crowded with osteons and other features, the bone area depicted in this image appears as an “open field,” relatively free of complex microstructural elements like densely packed osteons. This lack of structural complexity may allow for more extensive deformation, as there are fewer microstructural barriers to impede the propagation of the deformation. In regions that appear to be dominated by lamellar structures with few osteons, the propagation of such deformations could be facilitated by the reduced mechanical impedance, allowing the viscoelastic deformation to travel farther from the indentation site.

The extended deformation length could indicate that the less dense microstructural areas of the bone are more susceptible to deformation under the same loading conditions. Since the bone’s mechanical properties are anisotropic and dependent on its microstructure, areas with lower hardness values may allow for energy to be released over a more extended area, resulting in a longer “screw-like” deformation path.

This observation could provide valuable insights into the behavior of bone under load, especially in understanding how different regions of bone may respond differently to mechanical stresses based on their hardness and microstructural characteristics. It may also have implications for the design of implants or the study of bone diseases that affect microstructural integrity.

Figure 8 shows (a) a scanning electron microscopy (SEM) image and (b) a macroscopic photograph. Both capture the distinctive “screw-like” deformation pattern manifested in bone tissue following a Vickers hardness test. This characteristic pattern offers a unique insight into the material’s response to localized indentation stress. The white arrows in these images point to a deflection in the “screw-like” deformation that appears to correlate with a cement line within the bone structure. Cement lines are known to be regions within the bone that can impact the mechanical response to stress [10]. They often serve as interfaces between different structural components of the bone, such as between osteons and interstitial bone. Cement lines play a crucial role in bone tissue’s mechanical integrity and toughness by acting as interfaces that can absorb energy and mitigate damage [10]. They contribute to bone toughness when the “screw-like” deformation encounters a cement line and part of the energy from the indentation load is redirected and used to change the deformation’s path.

This energy consumption due to deflection benefits the bone, as it increases toughness—a material’s ability to absorb energy and plastically deform without fracturing. By redirecting the deformation path, the cement line effectively blunts the deformation, slowing its progression and reducing its length. This process is critical to the bone’s ability to resist fractures, allowing it to handle the stresses of daily activities without sustaining significant damage. It is a remarkable adaptation that showcases how the microarchitecture of bone is optimized for its mechanical environment.

The deformation pattern interacts with the bone’s microstructural components, specifically osteons. Osteons are the fundamental structural units in compact bone, consisting of concentric layers of calcified matrix, and are critical in defining the mechanical properties of bone. The deflection of the deformation pattern by the osteon (white arrow) suggests that the bone’s microstructure, particularly the arrangement and orientation of osteons, significantly influences how bone responds to mechanical stress. This insight is pivotal because it indicates that the deformation mechanism is not only time-dependent, as evidenced by its relation to the dwell time of the indentation, but also dependent on the specific characteristics of the bone’s microstructure.

In medical and biomechanical engineering applications, recognizing that bone’s mechanical response depends on its microarchitecture is crucial. It informs the design of orthopedic implants, prosthetics, and strategies for bone disease treatment, ensuring that they accommodate the bone’s intrinsic ability to resist and dissipate mechanical stresses effectively. This understanding could lead to the development of tailored treatments that address specific areas within the bone, considering the unique mechanical environment influenced by its microstructural composition.

## 7. Conclusions

This study uncovered a novel viscoelastic deformation mechanism during the investigation of Vickers hardness of bone. It dived into the viscoelastic deformation responses of bone when subjected to mechanical stress from Vickers hardness testing. High-resolution scanning electron microscopy (SEM) was used to examine a distinct deformation pattern that emerges from the indentation site within the bone matrix, which has been informally described as “screw-like” due to its resemblance to a screw thread when viewed under an optical microscope.

The key conclusions and findings from the study can be summarized as follows:Unique Deformation Pattern: The consistent “screw-like” deformations noted in this study indicate a specific and reproducible response of bone’s microstructure to mechanical stress, revealing an intricate, directional, and controlled manner in which bone dissipates energy from the indentation force.Microstructural Influence: The pattern of deformation does not occur universally; it manifests in particular samples and specific regions, suggesting that the phenomenon is highly dependent on local microstructural characteristics. While the exact features triggering this deformation are not yet identified, this study clearly links this occurrence to the microscopic structure of the bone.Directionality and Repeatability: The direction and repeatability of the deformation imply that bone can manage the energy from mechanical stress in a predictable manner, which is crucial for its functionality as it must withstand varied and repeated loads without failure.Viscoelastic Nature of Bone: This study provides quantitative evidence of bone’s viscoelastic behavior by demonstrating a direct correlation between the length of the “screw-like” deformation and the duration of the indentation dwell time.Cement Line Interaction: Cement lines within the bone appear to alter the mechanical response to stress, contributing to the bone’s toughness by redirecting and consuming energy, thus increasing the bone’s ability to avoid fractures by blunting the progression of deformations.Clinical and Biomedical Relevance: Recognizing that bone’s mechanical response is contingent on its microarchitecture informs the design of orthopedic implants and strategies for bone disease treatment, ensuring that they accommodate bone’s intrinsic ability to resist and dissipate stresses effectively.

## Figures and Tables

**Figure 1 jfb-15-00087-f001:**
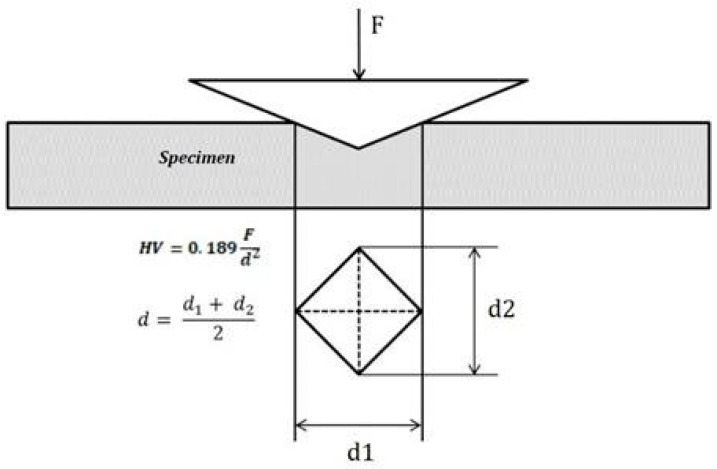
Vickers hardness test [22].

**Figure 2 jfb-15-00087-f002:**
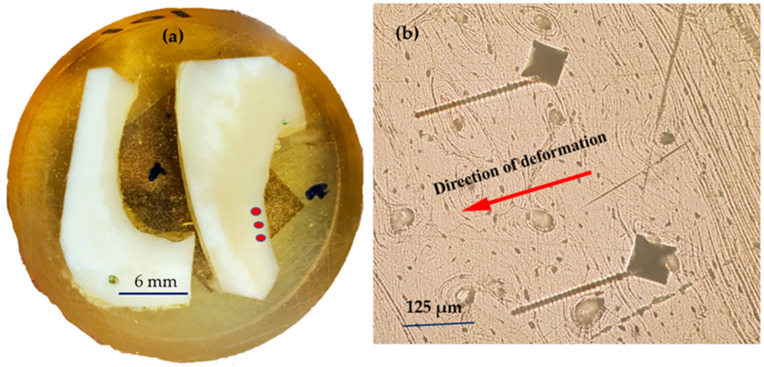
(**a**) Hardness test sites on deer tibia: a cross-sectional analysis; (**b**) microscopic detail of induced “screw-like” deformation from Vickers hardness testing.

**Figure 3 jfb-15-00087-f003:**
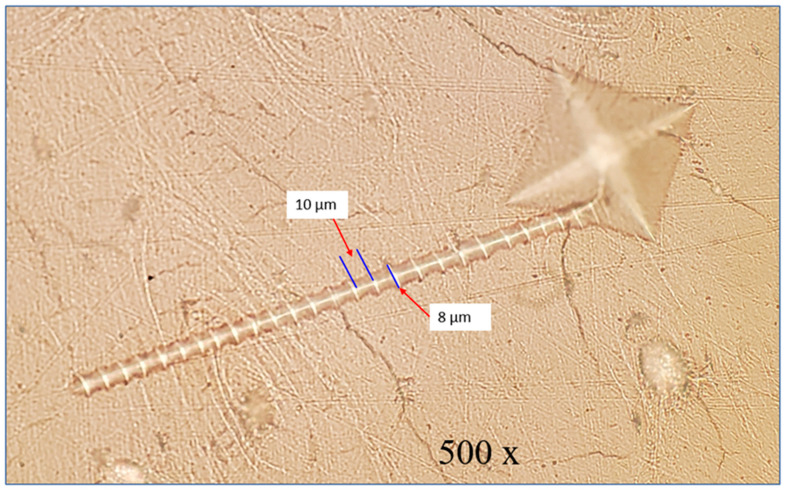
Vickers hardness indentation produced “Bone Deformation” seen in optical microscope as a screw thread.

**Figure 4 jfb-15-00087-f004:**
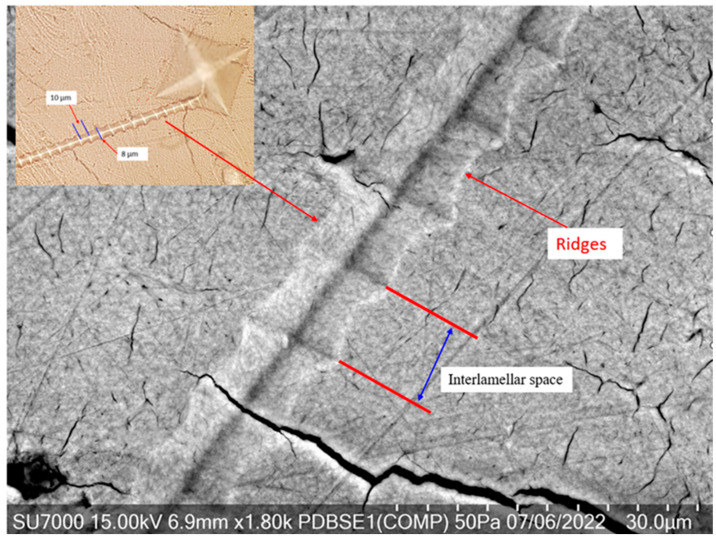
High-resolution SEM shows the screw-like deformation seen in Figure 3.

**Figure 5 jfb-15-00087-f005:**
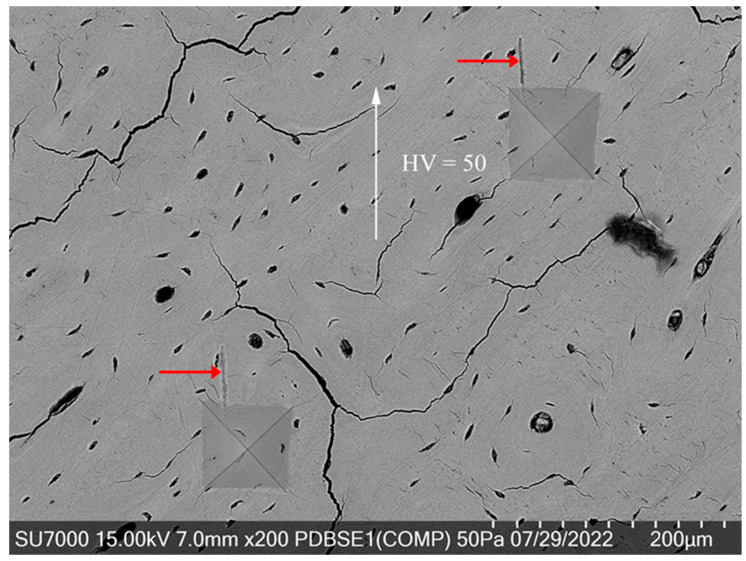
Consistent viscoelastic dynamics in proximal indentations.

**Figure 6 jfb-15-00087-f006:**
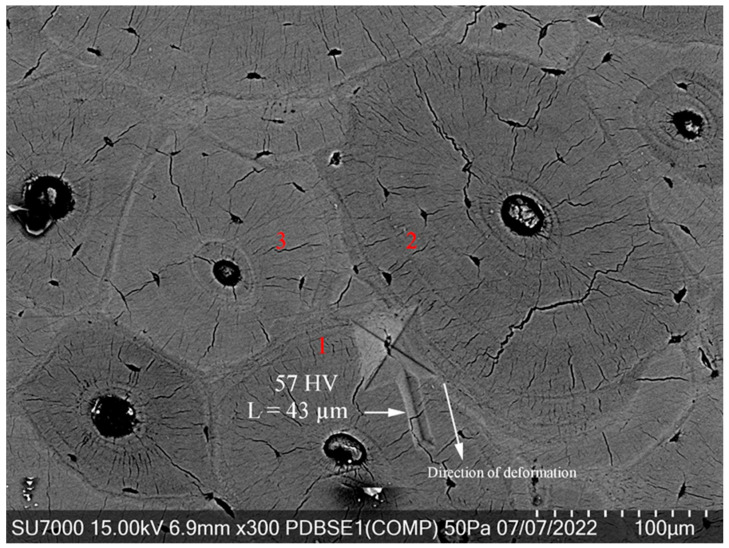
Viscoelastic deformation during Vickers indentation at osteon convergence.

**Figure 7 jfb-15-00087-f007:**
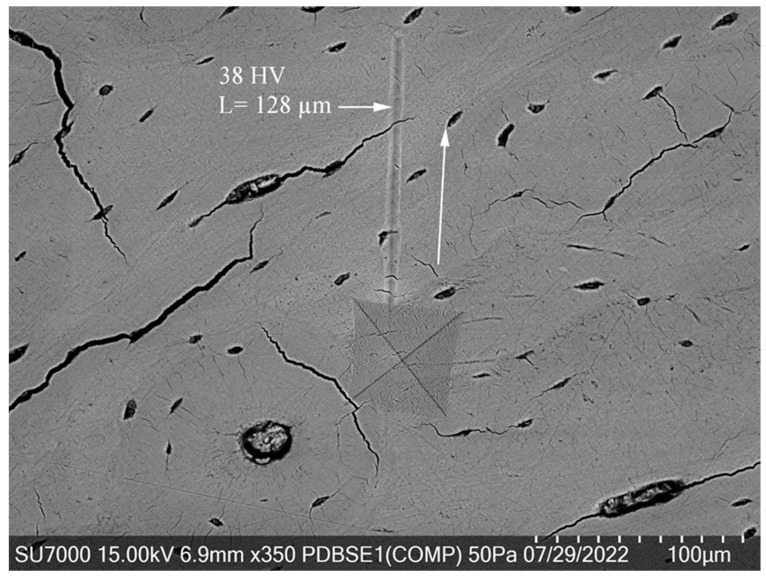
Extended viscoelastic deformation during Vickers indentation.

**Figure 8 jfb-15-00087-f008:**
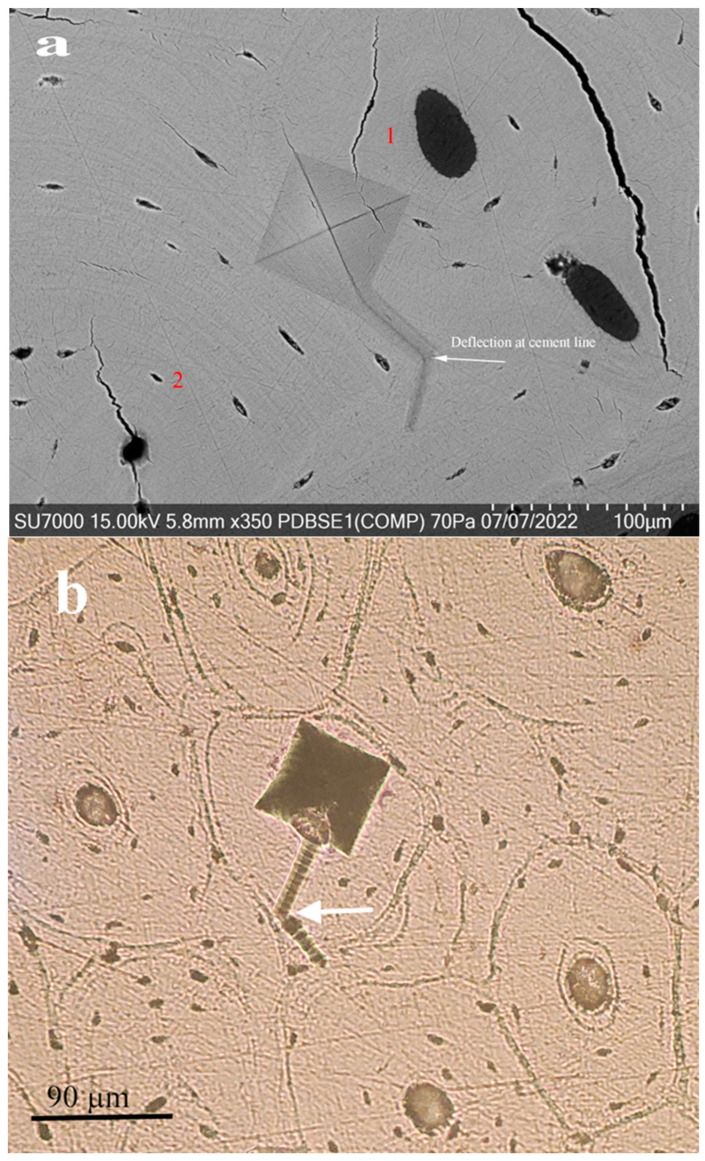
(**a**) SEM of “Screw-like” deformation deflected at the osteon cement lines. (**b**) Macroscopic image of “Screw-like” deformation deflected at the osteon cement lines.

## Data Availability

The original contributions presented in the study are included in the article, further inquiries can be directed to the corresponding author.

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
