# Peer review of "A Novel Viscoelastic Deformation Mechanism Uncovered during Vickers Hardness Study of Bone"

_jfb, 2024, doi:10.3390/jfb15040087_

Round 1

Reviewer 1 Report

Comments and Suggestions for Authors

Dear authors,

From the morphological suggestion of Vickers hardness, the viscoelastic deformation is considered, and the discussion is interested in Materials readers. The reviewer had some comments as following. Please take in consideration to revise this manuscript.

#1.     In Abstract, “Scanning Electron Microscopy” should change to lowercase.

#2.     Line 171 should be continuous before line 170.

#3.     Line 173, ASTM E384 should be referred to reference, maybe the number is 14.

#4.     Line 212-213, the caption of Figure 2 should change to lowercase except for first letter “Hardness”.

#5.     Line 240-242, 302, 326, 344, the underline should be removed if no meanings.

#6.     Line 396-, The authors suggested the ridge of screw-like pattern corresponded to that of collagen fibers in the bone. However, the orientation of collagen fiber could not show in this paper. The reviewer thought the orientation for collagen fiber are related to that for mineral, such as hydroxyapatite.

#7.     In Figure 5, the direction of two screw-like patterns is same. The authors explain why the screw-like pattern are same direction and can not be done in the opposite direction.

#8.     Bone have strongly anisotropic parts, such as long tubular bones. Those bone have discussed the bone quality. The authors should show more data and/or discuss in detail.

Comments on the Quality of English Language

One comment is that the first letter of a non proper noun should not be capitalized. Please check overall.

Reviewer 2 Report

Comments and Suggestions for Authors

The article A Novel Viscoelastic Deformation Mechanism Uncovered during Vickers Hardness Study of Bone explores a unique 'screw-like' deformation in bone under Vickers hardness indentation, utilizing high-resolution Scanning Electron Microscopy. It highlights the direct correlation between deformation length and indentation time, proving bone's viscoelastic properties. This research enhances understanding of bone's mechanical behavior, including energy dissipation and microstructural adaptation, contributing to the fields of material science and biomechanics. The findings are pivotal for future studies on bone's mechanical resilience and the development of biomimetic materials. The article is interesting and written generally correctly but will require minor corrections and additions before proceeding further. I have included detailed comments below.

Minor comments:

The introduction could use more information about bone structure please expand the first paragraphs of the paper. Authors will find useful information in publications:

https://doi.org/10.35784/jteme.538

https://doi.org/10.1007/978-1-59745-347-9_1

The purpose of the paper should be more clearly stated in the introduction so as to emphasize the importance of the research conducted.

Despite the long and correctly written discussion, the paper would have benefited from a summary and a brief description of the most important conclusions.

After corrections and additions are made, the work can be accepted for publication.

Comments on the Quality of English Language

Minor editing of English language required.

Reviewer 3 Report

Comments and Suggestions for Authors

The manuscript presents a pioneering investigation into a unique "screw-like" deformation mechanism observed in bone subjected to Vickers hardness indentation. The employment of advanced high-resolution Scanning Electron Microscopy (SEM) for the elucidation of microstructural alterations in bone induced by indentation stress is a convincing methodological choice, well-suited for the exploration of bone's viscoelastic properties.

The elucidation of a direct relationship between the extent of "screw-like" deformation and the duration of indentation dwell time unveils novel insights into the viscoelastic behavior of bone. Such insights bear significant implications for our comprehension of bone's fracture toughness and its resilience to mechanical perturbations.

The originality of the work is particularly underscored by the identification of the aforementioned deformation pattern within the bone under the specific conditions of indentation stress. This aspect of the research, notably emphasized in lines 244-247 by the authors, is commendable.

However, certain considerations regarding sample selection and treatment warrant attention. The heterogeneity inherent in biological specimens, such as variability in age and sex among “poor” deer samples, raises pertinent questions. Let me comment that as a 72 yo myself, I am well aware of bone characteristics with age. The inquiry into the number of independent samples analyzed, and the feasibility of conducting a statistical analysis to bolster the claimed correlations, is of essence. The absence of rigorous statistical validation emerges as a potential vulnerability, diminishing the impact of the methodological novelty presented.

Furthermore, the manuscript would benefit from the inclusion of control samples. Examination of bone samples that have not undergone Vickers hardness indentation could establish a foundational baseline for comparison with deformed specimens, thereby enhancing the understanding of bone's intrinsic microstructural and mechanical properties. The incorporation of synthetic materials or model composites, designed to emulate the microstructure and mechanical properties of bone, could serve as additional controls, facilitating meaningful comparisons.

An expanded discourse on the practical ramifications of these findings is desirable, particularly with respect to their relevance in the design of orthopedic implants, the management of osteoporosis, and the development of biomimetic materials.

The paper should maintain consistency in terminology for Vickers's test.

Considering these observations, I recommend that the Authors undertake a thorough revision of the manuscript to address the above concerns, which are likely to resonate with a broader readership.

Comments on the Quality of English Language

The paper should maintain consistency in terminology for Vickers's test.

Reviewer 4 Report

Comments and Suggestions for Authors

The present paper is very interesting and it will of interest to the readers of Journal of Functional Biomaterials. The research presented in the paper is novel and the paper is well-written; the introduction section of the paper provides an overview on the bone hardness and relevance of Vickers hardness test. Moreover, conducted methods are well described and obtained results are clearly presented. 

However, I would suggest to substitute Figure 1 which appears to be from Wikipedia/Wikimedia (Vickers hardness test - Wikipedia) with new figure. 

Round 2

Reviewer 1 Report

Comments and Suggestions for Authors

Dear authors,

The reviewer have read polite response and revised paper of yours. The reviewer is satisfied with the content of the revised paper. Finally, the authors should find the minor following points.

1. In Keywords, the uppercase or lowercase is corrected to "Bone Quality" or others after check the instruction.  

2. Is headline 5.1 necessary? In addition, check the capitalization of all headlines according to the instructions. For example, ”Screw-Like” would be more appropriate for ”Screw-like”, and "indentation" in 6.1.

3.  In line 399, a headline term, bold style" is required. 

4. In line 635-636, the sentence should be revised.

5. In line 649, "Microstructural Influence" should be bold.

Reviewer 3 Report

Comments and Suggestions for Authors

Authors have fulfilled all the requirements, the paper seems very improved congratulations to the Authors